# BDN: BLASCHKE DECOMPOSITION NETWORKS

## ABSTRACT

We introduce the Blashke Decomposition Network (BDN), a novel neural network architecture for analyzing continuous real-valued or complex-valued 1-D and 2-D signals - data types that existing architectures, such as transformers or recurrent networks, are not designed to model. These signals are common in medicine, biology, and other scientific domains, yet their analytic structure is often underutilized in machine learning. Our approach is based on the Blaschke decomposition, which "unwinds" a signal into a sequence of factors determined by its roots - the points in the complex unit disk where the analytic continuation of the signal vanishes. By iteratively peeling off these factors, the decomposition isolates oscillatory components of the signal and produces a compact representation. BDNs are trained to predict these roots directly, and we show that they provide powerful and interpretable representations for downstream tasks. We first design the architecture for 1-D signals and then extend it to 2-D using a wedge-based factorization, enabling the same framework to handle images and other spatially varying signals. Experiments on sensor-derived biomedical data, including electrocardiograms and phase holographic microscopy, show that BDNs achieve strong predictive performance while using fewer parameters than transformers, convolutional, or recurrent networks. Our code is available at: https://anonymous.4open.science/r/BDN-5603

## 1 INTRODUCTION

In the rapidly evolving landscape of neural networks, established paradigms have shown breakthrough performance on specific datatypes including convolutional neural networks (CNNs) for images, recurrent neural networks for discrete sequences (LeCun et al., 2015)(RNNs) (Hermans & Schrauwen, 2013; Cho et al., 2014), and transformer-based models for language (Vaswani et al., 2017). By contrast, continuous-valued oscillatory signals - ubiquitous in scientific and biomedical domains - have received far less architectural attention. These signals often exhibit phase modulation, and switch between periodic and aperiodic regimes, properties that are not naturally handled by existing discrete-sequence models. While most practitioners still use discrete-sequence models like LSTMs or transformers for these types of signals, there is currently no canonical architecture with inductive biases tailored to their analytic structure. Examples include electrocardiogram (ECG) recordings, gravitational waveforms, and phase microscopy, all of which exhibit oscillatory structure with varying phase, amplitude, and frequency content.

To address these challenges, we introduce the **Blaschke Decomposition Network (BDN)**, a new network architecture rooted in principles from complex and harmonic analysis. BDNs make use of the Blaschke unwinding series, which unwinds a signal step by step. At each stage, a root of the analytic continuation inside the unit disk is factored out, leaving a simpler residual to be decomposed further. This creates a product–sum structure: the signal is expressed as a weighted sum of products of Blaschke factors, where the factors are determined by the signal's roots and the weights by associated coefficients. Unlike Fourier or wavelet decompositions, which expand signals in terms of fixed basis functions, the Blaschke decomposition adapts to the signal itself. The representation is therefore compact, interpretable, and tied directly to the geometry of the signal's analytic continuation, with roots encoding oscillatory behavior and coefficients that weight their contribution.

BDNs turn the unwinding series into a fully differentiable neural framework by representing a signal as a truncated product–sum expansion, where each layer corresponds to factoring out a new Blaschke product. In this setup, the network learns both the roots, which capture oscillatory structure, and the coefficients, which weight their contribution. A learnable masking mechanism adap-

Figure 1: We propose Blaschke decomposition networks (BDNs) as a canonical architecture to model signals, especially complex-valued signals.

tively selects which roots are included in each factor. Stacking these layers produces a hierarchical, multi-resolution representation: coarse oscillatory structure is captured in the early stages, while finer details emerge in later ones. The result is an end-to-end trainable network that encodes signals directly through their learned root structure.

Finally, we extend the Blashke decomposition framework to 2-D signals through a wedge-based factorization. Here, the unit disk is partitioned into angular sectors, and Blaschke roots are assigned within each wedge to model oriented, scale-dependent features. This construction naturally generalizes the Blaschke decomposition from one-dimensional Hardy spaces to two dimensions (Rudin, 1969; 1980), enabling BDNs to capture spatial structure in images and microscopy data with the same principled root-based representation.

We evaluate BDN on both synthetic and real-world datasets. On synthetic signals composed of combinations of low- and high-frequency components, BDN learns root representations that separate these components cleanly. Using the learned roots as features, a simple classifier achieves near perfect accuracy in classifying the signal. Applied to electrocardiogram (ECG) data from the PTB-XL dataset (Wagner et al., 2020), BDN outperforms strong baselines including recurrent and transformer models, with higher sensitivity and specificity across multiple cardiac conditions. Beyond 1-D signals, we extend BDN to 2-D via the wedge factorization and test it on a digital holographic microscopy dataset of single-cell images. Here, BDN achieves the best classification accuracy across all three cell states - alive, apoptotic, and necroptotic - surpassing Fourier, wavelet, convolutional models. Together, these results highlight that BDN provides not only compact and interpretable representations through its learned roots, but also consistently strong predictive performance across a range of biomedical signal modalities.

In summary, the contributions of this work are threefold: (1) we introduce BDN, a new neural architecture that learns the Blaschke unwinding series for efficient and interpretable function representation; (2) we demonstrate that BDN captures complex dynamics more compactly than existing models, particularly high-frequency behaviors; and (3) we show that the internal learned components (roots and coefficients) serve as useful representations for classification and other downstream tasks. We validate these claims through a series of experiments on synthetic signals and real-world datasets.

## 2 BACKGROUND

In complex analysis, Blaschke products allow one to construct analytic functions on the unit disc $B(z)$ with zeros at some specified points $\{a_n\}_{n=1}^{\infty}$. They are employed in the canonical factorization theorem, which separates functions in Hardy spaces into distinct inner and outer components. The unwinding series, which leverages Blaschke products to represent $2\pi$-periodic real-valued signals, provides a powerful method for capturing harmonic components and decomposing signals. Together, these constructs offer a robust framework for analyzing and representing complex-valued functions.

### 2.1 BLASCHKE PRODUCTS AND HARDY SPACES

Blaschke products are instrumental in constructing analytic tools within Hardy spaces. A key component of these products is the Blaschke factor, which serves as the foundation for developing the Malmquist-Takenaka (MT) orthonormal bases in Hardy space. Through phase unwinding, these bases can be further adapted to form MT bases that align closely with the underlying signal properties. Additionally, the dynamics and composition of Blaschke products facilitate deeper insights into function structures, enabling enhanced representation and decomposition techniques.

**Analysis on the unit disk** A Blaschke product is an analytic function defined on the unit disk $\mathbb{D} = \{z \in \mathbb{C} : |z| < 1\}$ of the form

$$B(z) = z^m \prod_{n=1}^{N} \frac{|a_n|}{a_n} \frac{a_n - z}{1 - \bar{a}_n z} = z^m \prod_{n=1}^{N} b_{a_n}(z), \tag{1}$$

where $m$ is a non-negative integer and $\{a_n\} \subset \mathbb{D} \setminus \{0\}$ is a sequence of points such that $\sum_{n=1}^{\infty}(1 - |a_n|) < \infty$. Note that by construction, $B(z)$ is zero at each of the $a_n$ and also has a zero at the origin with mulitplicity $m$ (unless $m = 0$). Each term $b_a(z)$ is called a Blaschke factor associated with the zero $a$. For $a \neq 0$, the factor is given by $b_a(z) = \frac{|a|}{a} \frac{a-z}{1-\bar{a}z}$, while for $a = 0$ we adopt the convention $b_0(z) = z$. A zero of multiplicity $m$ at the origin therefore contributes a factor $z^m$ to the product.

The Hardy space $H^p(\mathbb{D})$, with $1 \leq p < \infty$, consists of all analytic functions $f$ on the unit disk $\mathbb{D}$ satisfying $\|f\|_{H^p(\mathbb{D})} := \sup_{0<r<1} \left( \frac{1}{2\pi} \int_0^{2\pi} |f(re^{i\theta})|^p \, d\theta \right)^{1/p} < \infty$. When $p = \infty$, the Hardy norm is defined as $\|f\|_{H^\infty(\mathbb{D})} := \sup_{z \in \mathbb{D}} |f(z)| < \infty$.

**Analysis on the upper half plane** Let $\{a_n\}_{n \geq 0}$ be a sequence (finite or not) of complex numbers with positive imaginary parts $\Im a_n$ and such that

$$\sum_{n \geq 0} \frac{\Im a_n}{1 + |a_n|^2} < \infty.$$

The corresponding Blaschke product is $B(z) = \prod_{n \geq 0} \frac{|1+a_n^2|}{1+a_n^2} \frac{z-a_n}{z-\bar{a}_n}$, where, $0/0$ is defined to be equal to 1 (which appears when $a_n = i$). The factors $\frac{|1+a_n^2|}{1+a_n^2}$ ensure the convergence of this product when there are infinitely many zeros.

Note that $H^p(\mathbb{D})$ can be identified with the set of $L^p$ functions on the torus $\mathbb{T} = \partial\mathbb{D}$ whose Fourier coefficients of negative order are equal to zero. We will alternate between analysis on the disk, and the parallel theory for analytic functions on the upper half plane $\mathbb{H} = \{x + iy \,|\, y > 0\}$. The space of analytic functions $f$ on $\mathbb{H}$ such that

$$\sup_{y>0} \|f(\cdot + iy)\|_{L^p(\mathbb{R})} < \infty$$

is denoted by $H^p(\mathbb{R})$. These functions have boundary values in $L^p(\mathbb{R})$ when $p \geq 1$.

### 2.2 The Unwinding Series

The unwinding series (Nahon, 2000; Coifman & Steinerberger, 2017; Coifman et al., 2017) is a construction for analyzing complex signals which may be derived using the canonical factorization theorem stated below.

**Theorem 1** (Canonical Factorization Theorem (Farnham, 2020))**.** *Let $F \in H^p(\mathbb{D})$ and $F \neq 0$. Then $F$ may be factored as:*

$$F = B \cdot F_1, \tag{2}$$

*where $B(z)$ is a Blaschke product accounting for the zeros of $F$ in $\mathbb{D}$, $F_1(z) \in H^p(\mathbb{D})$ is a function with no roots in $\mathbb{D}$, and $\|F\|_{H^p} = \|F_1\|_{H^p}$.*

To understand this theorem, we note that restricted to the boundary of the unit disk, $|B(z)| = 1$, which implies that $|F_1(z)| = |F(z)|$ and gives rise to the analogy that $B$ is the "frequency" of the signal $F$ while $F_1$ is its "amplitude".

To construct the unwinding series, we first write

$$F(z) = F(0) + \big(F(z) - F(0)\big). \tag{3}$$

Then, since the second term, $F(z) - F(0)$, has at least one root at the origin, it admits a non-trivial Blaschke factorization, as presented in Theorem 1. Therefore, we may iteratively apply Theorem 1

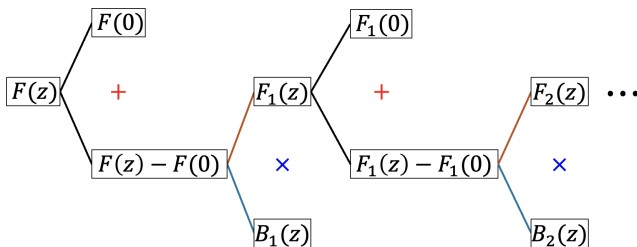

Figure 2: Graphical illustration of Blaschke unwinding process.

to write

$$
\begin{aligned}
F(z) &= F(0) + (F(z) - F(0)) \\
&= F(0) + B_1(z)F_1(z) \\
&= F(0) + B_1(z)(F_1(0) + (F_1(z) - F_1(0))) \\
&= F(0) + F_1(0)B_1(z) + F_2(0)B_1(z)B_2(z) + F_3(0)B_1(z)B_2(z)B_3(z)\dots \\
&= F(0) + \sum_{k \geq 1} F_k(0) \prod_{j=1}^{k} B_j(z).
\end{aligned}
\tag{4}
$$

This iterative process is visualized in Figure 2. The result, referred to as the *unwinding series*, is an orthogonal expansion for a function $F \in H^p(\mathbb{D})$ based on the idea that, in the repeated factorizations $F_k(z) - F_k(0) = B_{k+1} \cdot F_{k+1}$, each function $F_{k+1}$ is "simpler" than the corresponding $F_k$ because the winding number around the origin decreases. The phase of $B$ is considered as the primary object of study due to its expressive capabilities. As evidence of the expressive power of the Blaschke products, it was noted in Coifman & Peyri that each Blaschke term can be expressed as $B(z) = \exp(2i\theta(z))$, where $\theta$ is the phase for the function $B$. We present this formally as the following proposition. For a proof, please see Appendix A.1.

**Proposition 1.** *Let $\{a_n\}_{1 \leq n \leq N}$ be a sequence of complex numbers in the upper half-plane, i.e., $a_n = \alpha_n + i\beta_n$ where $\alpha_n, \beta_n \in \mathbb{R}$ and $\beta_n > 0$. Then the Blaschke product on the line*

$$
\prod_{n=1}^{N} \frac{z - a_n}{z - \bar{a}_n} = \exp(2i\theta(z)),
\tag{5}
$$

*where $\theta(z) = \sum_{n=1}^{N} \sigma(\frac{z - \alpha_n}{\beta_n})$, $\sigma(z) = \arctan(x) + \pi/2$.*

This phase is equivalent to the construction of a standard neuron in a neural network where $\sigma$ is taken to be the activation function of the neuron, each $\alpha_n$ is a bias term and each $\beta_n$ is a weight. This connection with neural networks demonstrates the versatility and expressivity of the method for generating highly complex functional representations.

Traditionally, the Blaschke decomposition is accomplished through the use of the deterministic algorithm of Guido and Mary Weiss (Weiss & Weiss, 1962). This series converges exponentially as observed in Nahon (2000); Coifman & Peyrière (2021) and has stability under white noise (Coifman et al., 2017).

From Proposition 1 and Eq. 4, the finite unwinding series can be expressed as

$$
\hat{f}(z) = \Re(F(z)); \quad F(z) = \sum_{\ell=1}^{L} c_l \prod_{k=1}^{\ell} B_k(z); \quad B_k(z) = \exp(2i\theta_k(z)).
\tag{6}
$$

The convergence of Eq. (6) is provided in Theorem 2. For a proof, please see, Appendix A.1.

**Theorem 2** (Convergence, Nahon (2000)). *For all $f \in H^2(\mathbb{D})$ and $\varepsilon > 0$ there exists $\{B_k\}_{1 \leq k \leq L}$ and $\{c_l\}_{1 \leq l \leq L} \in \mathbb{C}$ such that, if $L$ is sufficiently large, the approximation in Eq. (6) satisfies*

$$
\left\| f(z) - \hat{f}(z) \right\|_2 < \varepsilon.
$$

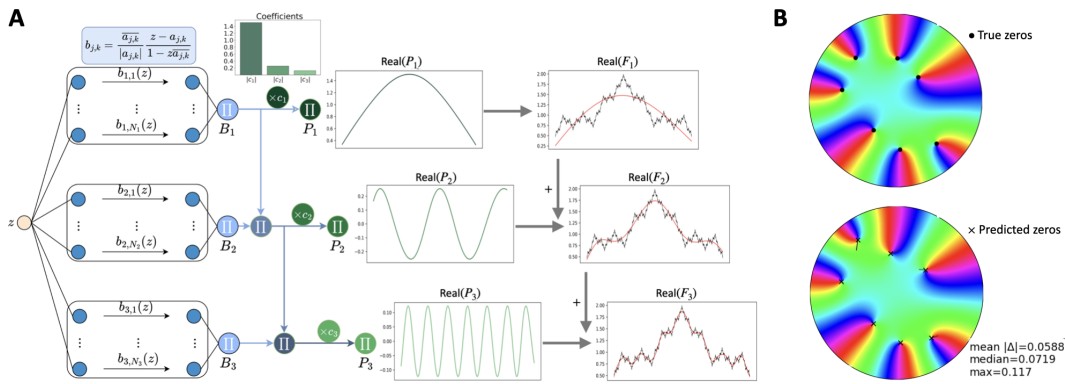

Figure 3: A. Learning Weierstrass function $f$ using 3-layer BDN: $\hat{f} = \text{Real}(F)$; $F = c_0 + c_1 B_1 + \cdots + c_3 B_1 B_2 B_3$. B. Comparison of the predicted and ground truth zeros (roots) in the unit disk.

## 3 BLASCHKE DECOMPOSITION NETWORK (BDN)

### 3.1 BDN ARCHITECTURE

To present our full BDN architecture, which also handles signals in higher dimensions, we begin with the one dimensional case, establishing it as the fundamental building block of the network. We then extend this foundation to two dimensional case, demonstrating how the wedge decomposition can be utilized to learn signals in higher dimensions.

**BDN as Univariate Functions** The BDN architecture is capable of approximating both *real- or complex-valued* analytic functions. In the case where the target function $F$ is complex-valued, BDN approximates $F$ via a truncated unwinding series modeled after Eqn. 4:

$$F_L = c_0 + c_1 B_1 + c_2 B_1 B_2 + \ldots + c_k B_1 B_2 \ldots B_k + \ldots + c_L B_1 B_2 \ldots B_L = \sum_{k=0}^{L} P_k, \quad (7)$$

where the $B_k$ are learnable Blaschke products with at most $N_k$ roots, and $c_k$ are learnable coefficients, analogues to the "frequency" and "amplitude" of a signal. We interpret each of these unwindings, i.e., factorizations $B_k F_k$, as a layer of our network. In the case where the target is a real-valued function $f$, we first learn a complex-valued function $F$ as in Eq. 7 and then take the real part, i.e., we set $f = \Re(f)$. We note that each term $P_k = c_k \prod_{j=1}^{k} B_j$ aims to represent the target function $F$ at progressively finer levels of detail, analogous to the multi-resolution representations derived from wavelet transforms (Jawerth & Sweldens, 1994). The parameter $L$ in Eq. (7), which controls the number of terms, can be chosen to be large to ensure a high degree of resolution or chosen to be small for computational efficiency.

The architecture of BDN is defined as follows. For each layer $k$, we first parameterize the phase via $\theta(z) = \sum_{n=1}^{N} \sigma(\frac{z - \alpha_n}{\beta_n})$ where $\sigma(z) = \arctan(x) + \pi/2$. Taking the exponential of this phase, $\exp(2i\theta(z))$, yields a Blaschke product $B_k$. Each $B_k$ is then incorporated into the recursive construction of Eq. (6) together with an additional set of scalar parameters $\{c_k\}_{k=0}^{L}$. The model parameters are optimized by minimizing the objective

$$\mathcal{L}_{\text{reconstruction}} = \left\| f - \sum_{\ell=1}^{L} c_\ell \prod_{k=1}^{\ell} B_k \right\|_2$$

which produces both the roots $\{r_n : r_n = \alpha_n + \beta_n i\}_n$ and coefficients $\{c_k\}_k$ that best approximate the target function $f$. Figure 3A illustrates the progressive approximation of the Weierstrass function, $f(x) = \sum_{n=0}^{\infty} 0.5^{10} \cos(3^{10} \pi x)$, via a 3-layer BDN. As the layer index $k$ increases, each component $P_k$ contributes successively finer-scale details to the signal. Figure 3B compares the learned zeros (roots) with the ground-truth ones, highlighting BDN's ability to faithfully capture structure through Blaschke roots.

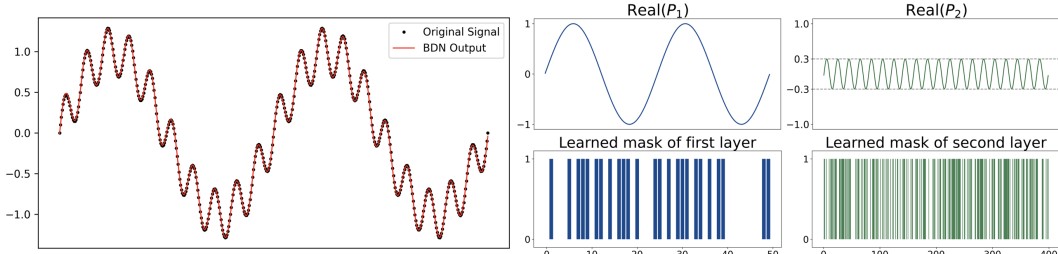

Figure 4: Learned components and masks of the function $f = \sin(4x) + 0.3\sin(40x)$ using a 2-layer BDN. In the first layer, 25 out of 50 roots are selected via the learned mask to form $P_1$, and in the second layer, 194 out of 400 roots are selected to form $P_2$.

**Mask Learning for Root Selections**  In each layer $k$ of BDN, the optimal number of roots needed to approximate the $k$-th component of the function is not known in advance. Rather than fixing this number as a hyperparameter, which can be both problem-dependent and costly to tune, we introduce a learnable binary mask to automatically select the most relevant roots for each layer. Specifically, the component function $\theta_k$ is defined as:

$$\theta_k(x) = \sum_{n=1}^{N_k} \chi_n \cdot \sigma\left(\frac{x - \alpha_n}{\beta_n}\right) \tag{8}$$

where $\chi_n$ ($= 0$ or $1$) is the learnable mask that determines whether the $n$-th root contributes to the approximation. This mechanism allows the network to adaptively choose how many and which roots to use, effectively pruning unnecessary components and reducing overfitting. As a result, the model remains robust across different choices of $N_k$, the maximum number of roots allowed per layer, and can efficiently allocate capacity where it is most needed. Figure 4 illustrates a two-layer BDN approximating a signal composed of two distinct frequency components, along with the learned masks used to select the appropriate number of roots in each layer. The network accurately captures the corresponding frequency content and coefficients at each scale, resulting in a near-perfect reconstruction of the original signal.

**Computational Complexity**  The BDN architecture is organized into $L$ layers of the unwinding series. At layer $l$, the model learns $l$ Blaschke products, and each product contains $P$ learnable Blaschke factors. Thus the total number of learnable factors across all layers is $\sum_{l=1}^{L} l \cdot P = P \cdot \frac{L(L+1)}{2} = O(L^2 P)$. Since each factor is parameterized by a root and a coefficient, the overall parameter count scales quadratically with depth $L$ and linearly with number of factors $P$. This product-sum structure allows BDN to capture increasingly fine-scale behavior while keeping the parameter growth controlled.

### 3.2 Wedge Decomposition for Image Representations

Extending Blaschke decomposition from one-dimensional signals to images requires capturing structure along both spatial dimensions. To this end, we introduce a *wedge decomposition*, which partitions the complex unit disk into angular sectors (wedges) and assigns Blaschke roots within each wedge to model oriented, scale-dependent features. Formally, let $\mathbb{D} = \{z \in \mathbb{C} : |z| < 1\}$ denote the unit disk, and divide it into $M$ disjoint wedges

$$W_m = \{z \in \mathbb{D} : \theta_m \le \arg(z) < \theta_{m+1}\}, \quad m = 1, \dots, M, \tag{9}$$

with angular boundaries $0 = \theta_1 < \theta_2 < \cdots < \theta_{M+1} = 2\pi$. For an image channel $I : \mathbb{R}^2 \to \mathbb{R}$, we first construct its analytic extension $F : \mathbb{D} \to \mathbb{C}$ using the complexification procedure (Hilbert transform followed by Poisson extension; see Appendix A.2). This step maps the spatial domain of the image into the unit disk, producing an analytic function that can be factorized using Blaschke products. In modalities such as digital holographic microscopy (DHM), however, the reconstructed hologram is already complex-valued, so the wedge decomposition can be applied directly to the phase image. Within each wedge $W_m$, we learn Blaschke factors

$$B_m(z) = \prod_{n=1}^{N_m} \frac{|a_{m,n}|}{a_{m,n}} \frac{a_{m,n} - z}{1 - \bar{a}_{m,n}z}, \quad a_{m,n} \in W_m, \tag{10}$$

where $N_m$ is the maximum number of roots allowed in wedge $W_m$. Analogous to mask learning for root selection in the univariate case, we introduce a gating variable $\chi_{m,n} \in \{0, 1\}$ to select the most relevant roots. The full wedge-decomposed Blaschke representation is then

$$F(z) \approx \sum_{\ell=0}^{L} c_\ell \prod_{k=1}^{\ell} \left( \prod_{m=1}^{M} \prod_{n=1}^{N_m} B_{m,n}(z) \, \chi_{m,n} \right). \tag{11}$$

Intuitively, roots near the origin correspond to coarse-scale isotropic features, while roots closer to the boundary encode fine-scale oscillations. By distributing roots across angular wedges, the network learns oriented features (such as edges and textures) at multiple scales. This is analogous to steerable filters in CNNs, but here the decomposition arises directly from the analytic structure of Hardy space functions.

## 4 RESULTS

### 4.1 BLASCHKE ROOTS FOR 1-D SIGNAL CLASSIFICATION

We demonstrate how the roots learned by BDN during training can be used as features for classification tasks on both synthetic and ECG signals, highlighting the method's versatility across different data modalities. In each case, BDN is used to fit the signals, and the extracted roots serve as input features for separate classifiers.

**Toy Signals**   We construct synthetic signals using combinations of **first-tier** and **second-tier** frequencies, defined as

$$f_{k,j}(x) = \sin(k\pi x) + 0.5 \sin(10j\pi x) + p,$$

where $k \in \{1, \ldots, 5\}$, $j \in \{1, \ldots, 5\}$, and $p \sim \mathcal{N}(0, 0.1)$ is Gaussian noise. This yields 25 distinct classes, corresponding to all combinations of first- and second-tier frequencies. Each signal is fitted using a two-layer BDN of size [24, 256]. Figure 5 illustrates the learned BDN roots in the com-

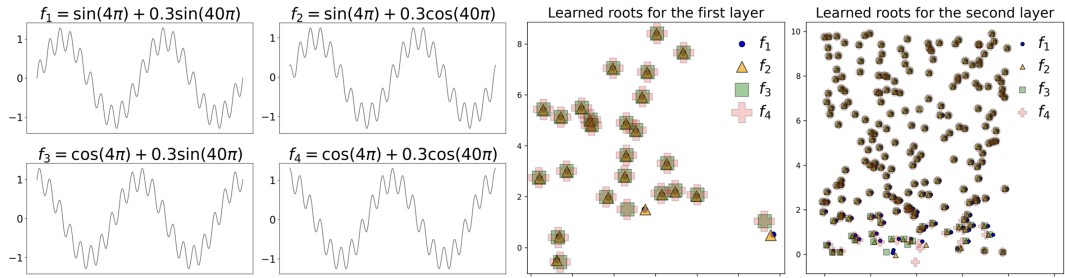

Figure 5: Learned roots of functions using BDN. Two layers with identical hyperparameters are used across all functions.

plex plane for four signals (without noise) that share identical first-tier and second-tier frequency components. Across different functions and layers, the roots consistently locate in similar locations, suggesting that the BDN learns a stable and distinctive representation for common frequency structures. A three-layer MLP classifier trained on the extracted roots achieves **99% accuracy** for first-tier frequencies, **99.9%** for second-tier frequencies, and **95.5%** when classifying all 25 combinations jointly.

**ECG Signals**   We use the PTB-XL dataset (Wagner et al., 2020), which contains 21,837 10-second 12-lead ECG recordings from 18,885 patients, each annotated by up to two cardiologists. The dataset includes normal ECGs and four types of abnormalities for multi-label classification of cardiovascular diseases: AD (Normal, 42.4%), MI (Myocardial Infarction, 25.6%), STTC (ST/T changes, 24.5%), CD (Conduction Disturbance, 22.9%), and HYP (Hypertrophy, 12.4%). Following the paradigm of Narotamo et al. (2024), we used only three leads (I, II, and V2) which provide complementary views of the heart's electrical activity across different planes: Lead I captures the horizontal plane between the arms, Lead II captures the inferior axis from right arm to left leg, and Lead V2 records anterior

chest activity. Using this setup, 17,111 recordings were used for training, 2,156 for validation, and 2,163 for testing, reflecting the class imbalance with AD as the largest and HYP as the smallest category. For each subject, a three-layer, one-input (time), three-output (three leads) BDN is fitted to their ECG signals. The roots extracted from BDN are then used as features for a three-layer MLP classifier for multi-class classification.

Table 1: Classification results obtained with different models. The best results are highlighted in bold and second best are underlined. *Spec*, *Sens*, and **G-Mean** denote specificity, sensitivity, and the geometric mean of these two values, respectively.

| Models | MI | | STTC | | CD | | HYP | | AD | | Total | | G-Mean |
|---|---|---|---|---|---|---|---|---|---|---|---|---|---|
| | Sens | Spec | Sens | Spec | Sens | Spec | Sens | Spec | Sens | Spec | Sens | Spec | |
| GRU (Cho et al., 2014) | 84.81 | 78.82 | 83.37 | 83.96 | 78.71 | 86.31 | 72.24 | 71.74 | 77.11 | 87.20 | 79.67 | 81.04 | 80.35 |
| LSTM (Hochreiter & Schmidhuber, 1997) | 73.60 | 85.90 | 82.79 | 84.39 | 82.12 | 78.68 | 76.81 | 65.63 | 68.24 | 90.80 | 75.42 | 80.09 | 77.72 |
| BiGRU (Narotamo et al., 2024) | 79.20 | 82.24 | 82.41 | 83.78 | 80.12 | 87.93 | 76.43 | 67.05 | 76.89 | 88.16 | 78.95 | 81.07 | 80.00 |
| BiLSTM (Narotamo et al., 2024) | 78.84 | 82.17 | 84.89 | 80.79 | 78.71 | 87.33 | 71.10 | 70.05 | 75.68 | 87.12 | 78.18 | 80.87 | 79.51 |
| CNN+GRU (Narotamo et al., 2024) | 72.88 | 78.88 | 84.13 | 75.30 | 77.47 | 85.89 | 66.54 | 70.63 | 69.55 | 86.16 | 72.33 | 78.35 | 75.28 |
| CNN+LSTM (Narotamo et al., 2024) | 71.61 | 77.70 | 88.15 | 71.95 | 68.67 | 84.98 | 63.50 | 68.84 | 73.38 | 84.88 | 74.04 | 77.06 | 75.53 |
| GRUAtt (Narotamo et al., 2024) | 83.72 | 80.50 | 80.88 | 86.16 | 76.71 | 85.89 | 78.33 | 63.53 | 69.11 | 89.44 | 76.55 | 80.15 | 78.33 |
| ECGTransForm (El-Ghaish & Eldele, 2024) | 77.84 | 62.65 | 88.46 | 75.84 | 71.66 | 88.38 | 69.88 | 71.22 | 78.59 | 84.73 | 77.29 | 76.56 | 77.14 |
| **BDN+MLP (ours)** | 85.26 | 81.42 | 87.29 | 85.73 | 85.98 | 88.71 | 80.04 | 75.62 | 74.22 | 89.55 | **82.56** | **84.20** | **83.37** |

To assess model performance, we computed several standard metrics. *Sensitivity* (or recall) measures the ability of the model to correctly identify positive cases of each disease and is defined as $Sens = TP/(TP + FN)$, where $TP$ and $FN$ are the numbers of true positives and false negatives, respectively. *Specificity* quantifies the ability to correctly identify negative cases (healthy subjects) and is given by $Spec = TN/(TN + FP)$, where $TN$ and $FP$ are the numbers of true negatives and false positives. The *geometric mean* (*G-Mean*) combines sensitivity and specificity to provide a balanced measure across classes, ensuring that under-represented classes are not overshadowed by dominant ones: $G\text{-}Mean = \sqrt{Sens \times Spec}$. Decision thresholds for the sigmoid outputs were optimized individually for each disease based on the highest *G-Mean* on the validation set, and the epoch with the lowest validation loss was used to extract the final results.

Table 1 summarizes the comparison between our model and the state-of-the-art models from (Narotamo et al., 2024), reporting *Sens*, *Spec*, and *G-Mean* for each class. Our method consistently outperform existing state of the art models, demonstrating the versatility of the learned representation from BDN.

### 4.2 Blaschke Roots for Holographic Microscopy Classification

Next we tested BDN on a Digital Holographic Microscopy (DHM) dataset. DHM provides a label-free and quantitative imaging modality by recording the interference pattern between a reference beam and the light scattered by biological specimens. The dataset introduced in Verduijn et al. (2021) consists of single-cell holograms of L929 fibroblast cells in three different states: alive (untreated), apoptotic (cells undergoing programmed cell death, e.g. in tissue homeostasis and cancer therapy), and necroptotic (cells undergoing a distinct form of regulated necrosis, often linked to inflammation and immune responses). Being able to discriminate these states is biologically important because apoptosis and necroptosis can look similar under a microscope but have very different implications for disease and treatment. Unlike traditional fluorescence assays, which require chemical labels that perturb the cells, DHM provides a non-invasive alternative that can scale to high-throughput drug screening and mechanistic studies of cell death.

The dataset contains 10,780 alive, 49,991 apoptotic, and 9,840 necroptotic single-cell holograms, with approximately 70,000 additional cells held out for evaluation. To prevent class imbalance during training, we randomly subsampled each class to match the size of the smallest class. We compared BDN against several baselines representative of classical and deep learning approaches. Fourier features were obtained by computing the 2D Fourier transform of each hologram, flattening the log-magnitude spectrum, and classifying with a multilayer perceptron (MLP). Wavelet features were extracted using a two-dimensional discrete wavelet transform (Daubechies-4 wavelet, three levels of decomposition). The resulting approximation and detail coefficients were concatenated into a fixed-length vector and classified with an MLP. As a deep learning baseline, we trained a ConvNet consisting of three convolutional layers (each followed by batch normalization and ReLU), a max pooling layer, and two fully connected layers. We also trained a convolutional autoencoder on

Table 2: Classification accuracy (%) on holographic microscopy dataset. Mean and standard deviation are computed across five runs. Best results in bold, second-best underlined.

| Model | Alive | Apoptotic | Necroptotic |
|---|---|---|---|
| Fourier+MLP | $78.5 \pm 0.7$ | $75.2 \pm 0.6$ | $79.1 \pm 0.8$ |
| Wavelet+MLP | $89.7 \pm 0.6$ | $83.7 \pm 0.6$ | $92.4 \pm 0.4$ |
| ConvNet | $82.1 \pm 0.6$ | $78.4 \pm 0.5$ | $84.2 \pm 0.7$ |
| ConvAE+MLP | $84.8 \pm 0.5$ | $80.2 \pm 0.6$ | $87.1 \pm 0.5$ |
| VGG-19 (finetuned) | $87.7 \pm 0.3$ | $81.6 \pm 0.4$ | $88.9 \pm 0.3$ |
| **BDN+MLP (ours)** | $\mathbf{92.3 \pm 0.4}$ | $\mathbf{88.2 \pm 0.5}$ | $\mathbf{95.6 \pm 0.3}$ |

the DHM dataset - the encoder comprised three convolutional layers with stride-2 downsampling, reducing each $66 \times 66$ hologram to a 128-dimensional latent vector, which was then fed into an MLP for classification. Finally, we evaluated VGG-19, which we fine-tuned in the same manner as described in Verduijn et al. (2021). For BDN, we applied a three-layer analytic factorization to each hologram and passed the extracted roots to a lightweight MLP classifier.

As shown in Table 2, BDN achieves the best performance across all three cell states, with particularly strong improvements for apoptotic cells, which are the most difficult to classify. On average, BDN reaches over $90\%$ accuracy with low variance across training runs, outperforming both feature-based and deep learning baselines.

## 5 RELATED WORK

Deep learning has often drawn on classical signal processing to model oscillatory or multiscale structure. Scattering networks (Mallat, 2012) use wavelet cascades to build stable multiresolution features, while Fourier neural operators (Li et al., 2021) and other spectral methods (Kovachki et al., 2023) employ frequency-domain transforms to learn dynamics in physical systems. These approaches, however, rely on fixed bases and offer limited interpretability. A parallel line of work investigates complex-valued neural networks (Trabelsi et al., 2018), enabling phase-aware modeling but typically without analytic guarantees or structured decompositions. Our work differs from these by integrating the Blaschke unwinding series into a differentiable, learnable architecture. This design provides a principled frequency-domain factorization that adapts to data while yielding interpretable parameters, offering a new paradigm for modeling oscillatory signals and high-frequency dynamics.

## 6 DISCUSSION AND CONCLUSION

We have introduced the Blaschke Decomposition Network (BDN), a neural architecture that embeds principles from complex analysis into deep learning. By learning Blaschke roots and coefficients, BDN provides a structured frequency-domain decomposition that is both compact and interpretable. This approach enables efficient modeling of highly oscillatory signals, surpassing conventional architectures in tasks such as reconstruction and classification.

Beyond performance, the learned roots act as frequency selectors, analogous to the way biological sensory systems respond selectively to frequency bands in auditory or visual stimuli. This connection suggests opportunities for interdisciplinary research linking analytic decompositions, neural coding, and theories of hierarchical abstraction in neuroscience.

Going forward, several extensions are promising. One direction is to replace the discrete attention over Blaschke roots with a continuous, differentiable distribution, potentially improving gradient flow and generalization. Another is to extend BDN to handle spatially varying or spatiotemporal inputs, such as images and video, where localized oscillatory patterns play a key role. More broadly, BDN opens a pathway toward architectures explicitly designed for oscillatory and periodic data, offering both practical advantages and theoretical insight into frequency-based representations in machine learning.

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

## A  THEORY

### A.1  PROOFS

**Proposition 2.** *Let* $\{a_n\}_{1 \leq n \leq N}$ *be a sequence of complex numbers in the upper half-plane, i.e.,* $a_n = \alpha_n + i\beta_n$ *where* $\alpha_n, \beta_n \in \mathbb{R}$ *and* $\beta_n > 0$. *Then the Blaschke product on the line*

$$\prod_{n=1}^{N} \frac{z - a_n}{z - \bar{a}_n} = \exp(2i\theta(z)), \tag{12}$$

*where* $\theta(z) = \sum_{n=1}^{N} \sigma(\frac{z - \alpha_n}{\beta_n})$, $\sigma(z) = \arctan(z) + \pi/2$.

*Proof.* Since $\arctan(z) = \frac{1}{2i} \ln \frac{i-z}{i+z}$, we have that

$$\exp\left[2i \sum_{n=1}^{N} \sigma\left(\frac{z - \alpha_n}{\beta_n}\right)\right] = \exp\left[2i \sum_{n=1}^{N} \left(\frac{1}{2i} \ln \frac{i - (z - \alpha_n)/\beta_n}{i + (x - \alpha_n)/\beta_n} + \frac{\pi}{2}\right)\right]$$

$$= \prod_{n=1}^{N} (-1) \cdot \frac{i - (z - \alpha_n)/\beta_n}{i + (z - \alpha_n)/\beta_n}$$

$$= \prod_{n=1}^{N} \frac{z - (\alpha_n + \beta_n i)}{z - (\alpha_n - \beta_n i)}$$

$$= \prod_{n=1}^{N} \frac{z - a_n}{x - \bar{a}_n}.$$

$\square$

**Theorem 3** (Convergence). *Let* $f \in H^2(\mathbb{D})$. *For any* $\varepsilon > 0$, *there exists a sequence of Blaschke products* $\{B_k\}_{1 \leq k \leq L}$ *and coefficients* $\{c_l\}_{1 \leq l \leq L} \in \mathbb{C}$ *such that the approximation in Eq. (6) satisfies*

$$\left\| f(z) - \hat{f}(z) \right\|_2 < \varepsilon.$$

*Proof.* We will now prove that the approximation is an orthonormal decomposition and the convergence in norm of the residual goes to zero. Following the lead of Nahon (2000), we note that on the unit circle $\mathbb{T}$, the scalar product of two terms of $f$'s decomposition, $B_1 B_2 \ldots B_p$ and $B_1 B_2 \ldots B_n$, $n > p$ is:

$$\langle B_1 B_2 \ldots B_p, B_1 B_2 \ldots B_n \rangle = \oint_{\mathbb{T}} (\overline{B_1 B_2} \ldots \overline{B_p})(B_1 B_2 \ldots B_n) d\mu,$$

where $d\mu = \frac{dz}{z}$ is the measure derived by indentifying $\mathbb{T}$ with $[0, 2\pi]$ and writing $z = e^{i\theta}$.

Since $|B_k(z)| = 1$ when $|z| = 1$, this yields

$$\langle B_1 B_2 \ldots B_p, B_1 B_2 \ldots B_n \rangle = \frac{1}{2\pi i} \oint_{\mathbb{T}} B_{p+1} \ldots B_n \frac{dz}{z}.$$

Since the $B_k$ are analytic on $\mathbb{D}$, this implies

$$\langle B_1 B_2 \ldots B_p, B_1 B_2 \ldots B_n \rangle = 0$$

and so the terms $\{B_1 B_2 \ldots B_n\}_{n=1}^{L}$ are orthogonal. Since

$$\langle B_1 B_2 \ldots B_n, B_1 B_2 \ldots B_n \rangle = \frac{1}{2\pi i} \oint_{\mathbb{T}} |B_1 \ldots B_n|^2 \frac{dz}{z} = 1,$$

this implies

$$\|B_1 B_2 \ldots B_n\|_2 = 1,$$

thus normal. Therefore we have the orthonormality of the series. We now evaluate the residual part of the decomposition. The residual

$$\|f(z) - \hat{f}(z)\|_2 = \|c_{L+1} \prod_{k=1}^{L+1} B_k(z)\|_2$$

for some $c_{L+1}$. We have shown before the orthogonality of the decomposition it means that we can apply the Pythagorean theorem:

$$\|f\|_2^2 = \sum_{l=0}^{L} |c_l|^2 + \left\|c_{L+1} \prod_{k=1}^{L+1} B_k(z)\right\|_2^2 = \sum_{l=0}^{L} |c_l|^2 + |c_{L+1}|^2.$$

Hence for any $\epsilon > 0$, there must exist $c_0, c_1, ..., c_L$ such that $\|f\|_2^2 - \sum_{l=0}^{L} |c_l|^2 < \epsilon$. $\qquad\square$

### A.2 COMPLEXIFICATION OF A REAL SIGNAL

A bridge between analysis in Hardy Spaces and signal processing is given by the Hilbert transform and analytic projection using the Poisson kernel. A given input signal is assumed to be a $2\pi$ periodic function given by $s : [0, 2\pi] \to \mathbb{R}$ where we further assume $s \in L^2([0, 2\pi])$ and also identify periodic functions with functions on the unit circle $\mathbb{T}$. The circular Hilbert transform is a singular integral transform of $s$ denoted by $\mathcal{H}s$ and is given by:

$$\mathcal{H}s(\theta) := \frac{1}{2\pi}\text{p.v.} \int_0^{2\pi} s(t) \cot\left(\frac{\theta - t}{2}\right) dt, \tag{13}$$

where p.v. indicates that integral is defined in the principal-value sense (to avoid the singularity at $\theta = t$).

The Gabor complexification of a real signal $s$ is a complex-valued function $f \in L^2(\mathbb{T})$ given by:

$$f_s(e^{i\theta}) = s(\theta) + i\mathcal{H}s(\theta). \tag{14}$$

$f$ has a real component that agrees with $s$ and additionally satisfies $f \in L^2(\mathbb{T})$. It is known that there exists a unique analytic extension of $f$ to $\mathbb{D}$ which is achieved through circular convolution against the Poisson kernel $P_r(\theta)$ defined by:

$$P_r(\theta) := \frac{1 - r^2}{1 - 2r\cos(\theta) + r^2}. \tag{15}$$

The extension to the unit disk, $F_s$, is thus given by:

$$F_s(re^{i\theta}) := \frac{1}{2\pi} \int_0^{2\pi} P_r(\theta - t) f_s(e^{it}) dt, \quad 0 \le r < 1. \tag{16}$$

This extension is analytic in $\mathbb{D}$ and agrees a.e. on $\partial\mathbb{D} = \mathbb{T}$. This extension ensures that $F \in H^2(\mathbb{D})$, making it amenable to the techniques discussed above.

### A.3 EXTENSION TO MULTI-INPUT AND MULTI-OUTPUT MAPPINGS

The current BDN, as formulated in Eq. (7), is designed primarily for univariate functions and lacks the ability to adapt to the multi-input and multi-output scenarios. To this, we incorporate an **attention mechanism** applied to the phase parameter via

$$\Theta_k(\vec{z}) = \theta_k(\vec{z}) A_k, \tag{17}$$

where $\theta_k(\vec{z}) \in \mathbb{C}^{1 \times d_{\text{in}}}$ is the $d_{\text{in}}$-dimensional input phase and $A_k \in \mathbb{R}^{d_{\text{in}}, d_{\text{out}}}$ is an column normalized attention matrix corresponding to each Blaschke factor $k$. Here, $\Theta_k(\vec{z}) \in \mathbb{C}^{1, d_{\text{out}}}$ represents the phase after the attention mechanism. Consequently, the weighting phases $\theta$ enhance the expressiveness of the BDN by enabling it to learn weighted combinations of input dimensions tailored to each

output, thus accommodating the inherently multivariate nature of the target function. Hence, using Eq. (6) and (17), the BDN architecture formulation for the mapping case can be expressed as

$$\hat{f}(\vec{z}) = \Re(F(\vec{z})); \quad F(\vec{z}) = \sum_{\ell=1}^{L} c_l \prod_{k=1}^{\ell} B_k(\vec{z}) \quad \text{where} \quad B_k(\vec{z}) = \exp(2i\Theta_k(\vec{z})). \tag{18}$$

The convergence property of BDN extends naturally from univariate function learning to multi-input, multi-output mappings by incorporating an attention mechanism. Specifically, we consider each output component $f_q \in H^p(\partial\mathbb{B}_n)$, for $q \in \{1, 2, \ldots, d_{\text{out}}\}$, to admit a Blaschke unwinding series representation, where $\partial\mathbb{B}_n := \{z \in \mathbb{C}^n : |z| = 1\}$ denotes the boundary of the $n$-dimensional unit ball. This generalization relies on the mathematical foundation provided by the factorization theorems for Hardy spaces in several complex variables (Coifman et al., 1976).

**Theorem 4** (Coifman et al. (1976)). *Given $F$ in $H^1(\partial\mathbb{B}_n)$, there are $G_k, H_k \in H^1(\partial\mathbb{B}_n)$ such that*

$$F = \sum_{k=1}^{\infty} G_k H_k.$$

## B FUNCTION LEARNING

**Univariate Functions** We demonstrate how BDN effectively disentangles signal components through a learned unwinding series. Figure S1 shows an example of this series for the trigonometric polynomial $f = \sin(2x) + 0.6\sin(20x) + 0.2\sin(200x)$. The first term in the unwinding series isolates the primary oscillation, capturing the largest amplitude in the signal. Specifically, $c_1$ represents the dominant component, with its average modulus value centered around 1, while the phase of $P_1$ varies with a period of $\pi$ corresponding to $\sin(2x)$. This structure makes it easy to interpret the main oscillatory feature of the signal. The second term of the unwinding series reveals the finer oscillatory detail: $P_2$ oscillates with a shorter period of approximately $\pi/10$ while $c_2$ has a modulus value centered around 0.6, corresponding to the minor contribution from $0.6\sin(20x)$. The third terms reveals the finest oscillatory detail: $P_3$ oscillates with a period of approximately $\pi/100$ while $c_3$ has a modulus value centered around 0.2, corresponding to the least contribution from $0.2\sin(200x)$. This interpretability, enabled by the unwinding series structure in BDN, allows us to break down complex signals into meaningful, analyzable components, providing insights into the underlying harmonic structure of the data.

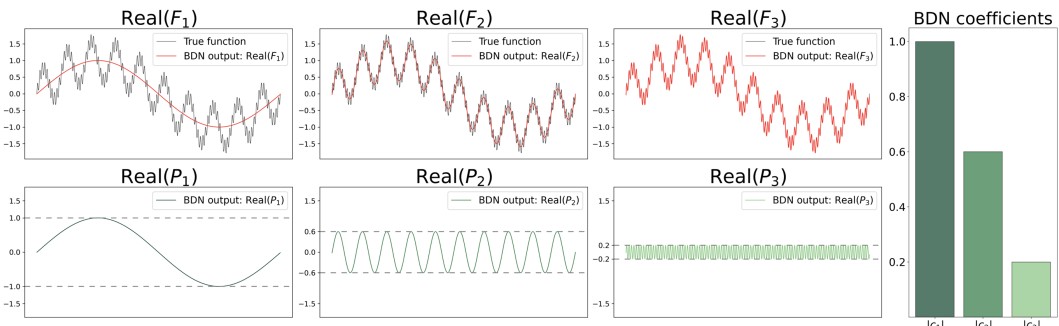

Figure S1: Learned unwinding components of the function $f(x) = \sin(2x) + 0.6\sin(20x) + 0.2\sin(200x)$ using a 3-layer BDN. The partial sums $F_i$ ($i = 1, 2, 3$) progressively approximate the target function at increasing levels of detail, while the corresponding Blaschke products $P_i$ capture the frequency components at each scale. BDN successfully reconstructs the original function with accurate coefficients and frequency components.

## C IMPLEMENTATION DETAILS

**Blaschke Product** The Blaschke products form the core of the BDN architecture, making their stability crucial during both the forward and backward passes. However, using the original formulation

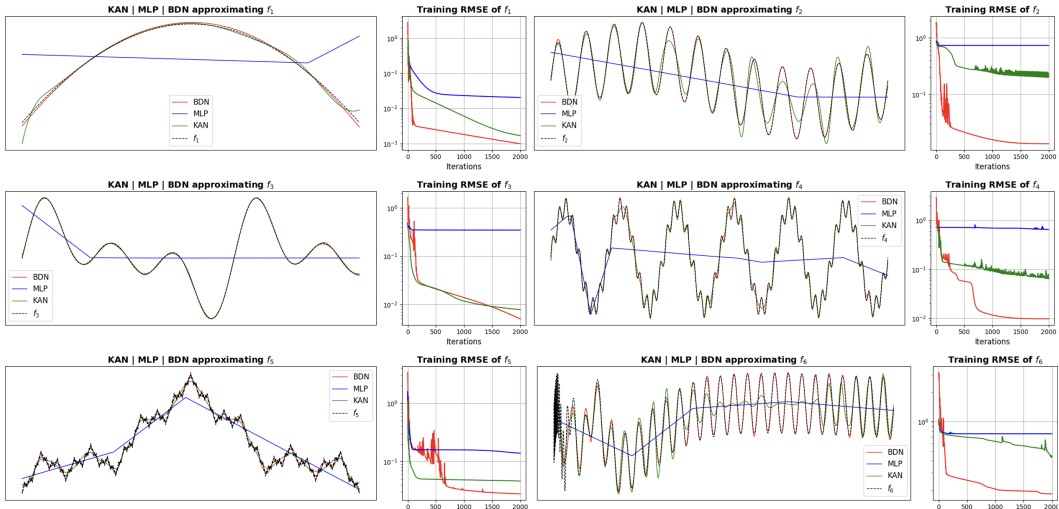

Figure S2: Comparison of univariate functions approximation using MLP, KAN, and BDN. BDN provides a closer fit to the ground truth function.

of the Blaschke product, as defined in Section 2, introduces numerical challenges. Specifically, the multiplicative nature of the product operator can lead to underflow and overflow issues, especially when handling values close to zero or very large magnitudes. These numerical instabilities impact the training dynamics and hinder effective gradient propagation. To address numerical instabilities, we reformulate the Blaschke product using Proposition 1, where the phase $\theta$ is computed as a sum before converting it into the complex domain with the Euler formula. This approach avoids the underflow and overflow issues associated with the original product form. Summing terms is numerically more stable than multiplying, and the Euler transformation preserves the complex structure of the Blaschke product without amplifying extreme values, enabling stable forward and backward passes in the BDN model.

**Mask Learning for Root Selection**  A key feature of the BDN architecture is its ability to adaptively select relevant Blaschke roots for signal representation. Each root in a BDN layer is associated with a learnable gating parameter which determines whether the corresponding root contributes to the output. To achieve differentiable selection, we employ a binary gating mechanism using a straight-through estimator (STE), where the forward pass applies a hard threshold to activate or deactivate a root, while the backward pass allows gradient flow as if the selection were continuous. Optionally, a Gumbel-Sigmoid relaxation can be used to introduce stochasticity in the root selection, enabling exploration during training. This mechanism allows the network to automatically identify and focus on the most informative components of the signal, effectively performing a form of mask learning over the root sections. The selected roots are then summed (or multiplied in the complex domain) to form the Blaschke product, providing a sparse yet expressive representation that captures the essential dynamics of the input signals.

**Parallelizing BDN**  BDN can achieve computational efficiency in distributed systems, as each $B_k$ can be computed independently. This parallelism enables significant speed-up by allowing concurrent processing of components across multiple nodes.

## C.1 APPLICATION IN PHYSICS

**Instantaneous Frequency of Gravitational Wave**  We extend the application of the Blaschke Decomposition Network (BDN) to the domain of gravitational wave modeling, with the goal of accurately capturing the intricate structure of gravitational waveforms. The BDN's inherent ability to represent complex analytic functions makes it particularly well-suited for this task. In addition to waveform reconstruction, we use the learned complex representation to compute the *Instantaneous*

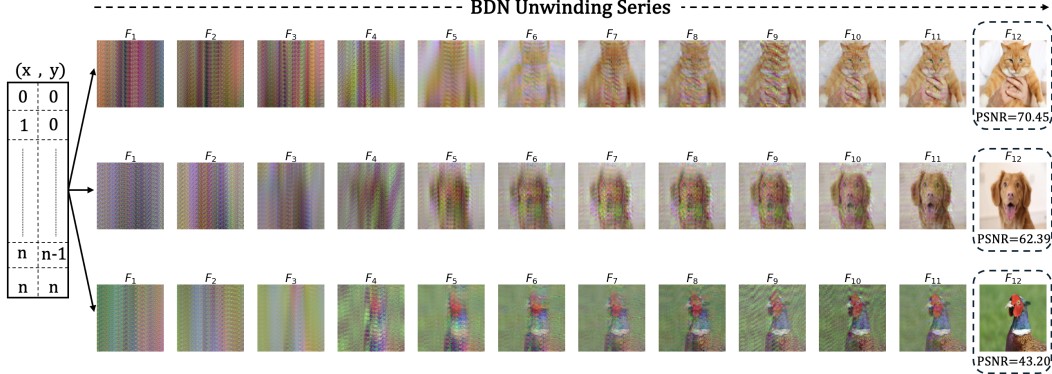

Figure S3: Image reconstruction task on images ($128 \times 128$ pixels). Each $F_i$ represents stages of the image learning process during the iterative Blaschke unwinding. Higher Peak Signal-to-Noise Ratio (PSNR) values indicate better reconstruction quality.

*Frequency (IF)*: a time-resolved feature that provides insight into the evolving frequency content of the signal.

Our experiments focus on the gravitational-wave event **GW150914**, the first direct observation of a binary black hole merger by the two detectors of the Advanced Laser Interferometer Gravitational-Wave Observatory (LIGO). The event was observed independently at the Hanford (H1) and Livingston (L1) sites, and its astrophysical origin has been extensively verified (Abbott et al., 2016). Each detector recorded a signal lasting 0.21 seconds, sampled at a high rate of 16,384 Hz, resulting in a finely resolved time series. The data are publicly available through the LIGO Open Science Center (`https://losc.ligo.org/events/GW150914/`), which also provides detailed documentation regarding the signal acquisition process and physical interpretation.

Because the gravitational wave signals are real-valued, we employ the *Hilbert transform* to construct their complex analytic extensions. Given a signal $s : [0, 2\pi] \to \mathbb{R}$ satisfying suitable regularity conditions, the Hilbert transform $\mathcal{H}(s)$ allows us to define a corresponding analytic signal $s^c \in H^p(\mathbb{D})$, where the real part matches $s$ on the boundary $\partial\mathbb{D}$. Specifically, we define:

$$s^c(e^{i\cdot}) = s(\cdot) + i\mathcal{H}(s(\cdot)),$$

where $\mathcal{H}$ denotes the Hilbert transform (see Appendix A.2 for details). An important advantage

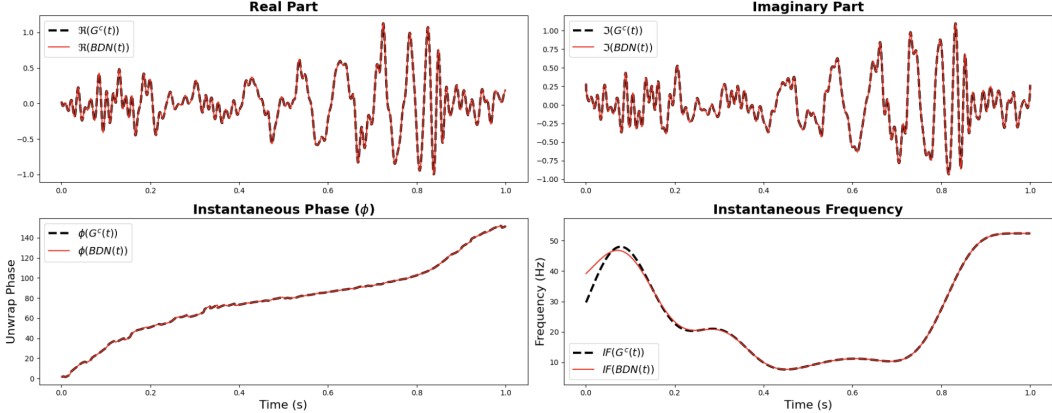

Figure S4: Ground Truth and BDN approximation to the complex gravitational wave signal for real, imaginary part instantaneous phase and IF.

of the BDN architecture is its theoretical ability to represent any analytic function—a property not shared by conventional models such as MLPs or KANs. To train the BDN on complex-valued data, we define a loss function that separately penalizes reconstruction errors in both the real and

imaginary parts:

$$\mathcal{L}(y, \hat{y}^c) = \mathcal{L}(\Re(y^c), \Re(\hat{y}^c)) + \mathcal{L}(\Im(y^c), \Im(\hat{y}^c)),$$

where $y^c$ is the complex ground truth and $\hat{y}^c$ is the BDN prediction.

Once a complex signal $s^c(t)$ has been learned, we compute its *Instantaneous Frequency* as:

$$IF(s^c(t)) = \frac{d\,\text{Phase}(s^c(t))}{dt}, \quad \text{with} \quad \text{Phase}(s^c(t)) = \arctan\left(\frac{\Im(s^c(t))}{\Re(s^c(t))}\right).$$

This formulation provides a window into the signal's dynamic frequency content—critical for analyzing gravitational waveforms that evolve rapidly in time due to the underlying astrophysical dynamics.

In our implementation, we used a 10-layer BDN architecture with a parameter setting of $\Lambda = 500$, optimized using the momentum-based optimizer proposed in Goyal (2017). As shown in Figure S4, the BDN model produces a highly accurate reconstruction of the complexified GW150914 signal, closely matching both the real and imaginary components. These results highlight the strength of BDN in learning smooth, oscillatory, and analytically structured signals. Its ability to recover the IF dynamics directly from data—without relying on explicit time-frequency transforms—demonstrates its utility as a tool for interpretable, data-driven analysis of gravitational wave events.

We further investigate the BDN's capacity for frequency analysis using synthetic chirp signals in Appendix C.2.

## C.2 FREQUENCY ANALYSIS

In Section C.1, we examine BDN's capability to learn analytical signals and extract meaningful frequency information through the phase component $\theta$ learned by the network. To validate the accuracy of BDN's instantaneous frequency predictions, we conduct experiments using a complex chirp signal defined as $Chirp(x) = \sin(2\pi(1 + 20x)x)$, with its analytical representation obtained through the Hilbert transform. As demonstrated in Figure S5, BDN accurately learns the analytical signal of the chirp function and precisely captures its increasing frequency pattern, as evidenced by the instantaneous frequency analysis.

This capability represents a significant advancement in signal processing applications. Unlike traditional methods that often require multiple preprocessing steps or face limitations with non-stationary signals, BDN provides a unified framework for simultaneously learning the signal representation and extracting frequency information. This integration is particularly valuable in real-world applications such as fault detection in rotating machinery, where varying frequency patterns can indicate equipment deterioration, or in biomedical signal processing, where frequency variations in ECG or EEG signals can reveal crucial diagnostic information. Furthermore, BDN's ability to handle non-stationary signals makes it especially suitable for analyzing complex phenomena in fields ranging from seismic data analysis to speech recognition, where traditional Fourier-based methods may fall short due to their inherent limitations with time-varying frequencies.

## C.3 IMAGE CONSTRUCTION INTERPRETATION

Figure S6 illusrates how a trained BDN allocates its capacity across spatial scale and orientation.

**Radial position $\rightarrow$ spatial scale** A single Blaschke factor is determined by its complex root $\zeta = r\,e^{i\theta}$ inside the unit disc. Writing the factor in the form $B_\zeta(z) = e^{i\phi}\frac{z-\zeta}{1-\bar{\zeta}z}$, one finds that the dominant Fourier component of $\arg B_\zeta$ has wavelength proportional to $(1-r)^{-1}$. The network learns roots with increasing modulus: $B_1$ and $B_2$ (top-left of the figure) have $r \ll 1$, so their phase varies slowly; from $B_3$ onwards the roots migrate towards the boundary, and the isophase stripes tighten until $B_{12}$ oscillates every few pixels. The ordering produced by SGD therefore tracks a low-to-high frequency progression predicted by the theory.

**Angular position $\rightarrow$ orientation** The argument $\theta$ of the root rotates the stripe pattern. BDN distributes the roots almost uniformly on the circle, producing a bank of direction-selective responses.

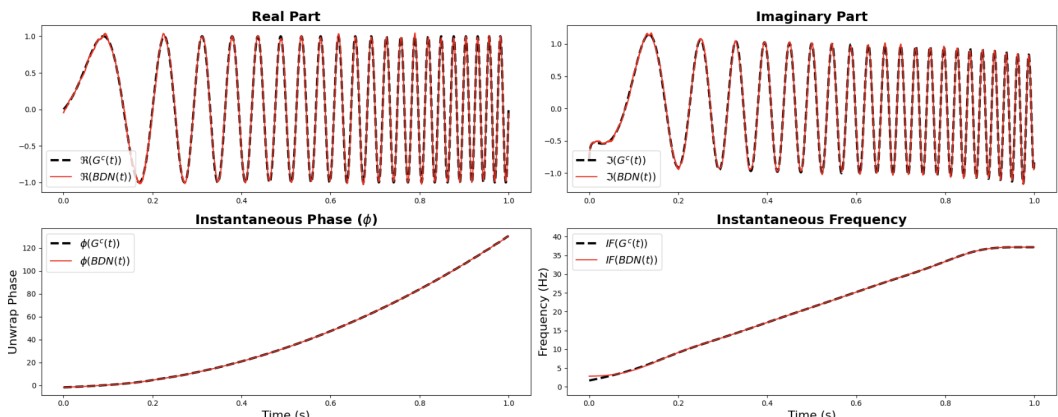

Figure S5: Ground Truth and BDN approximation to a complex Chirp wave signal for real, imaginary part instantaneous phase and IF.

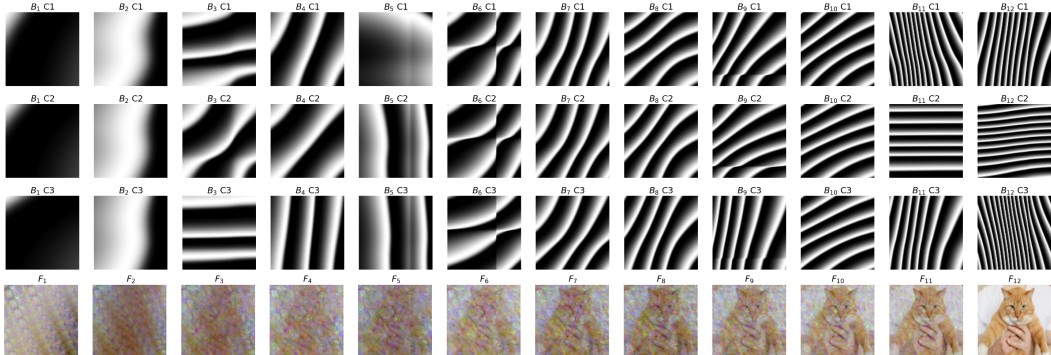

Figure S6: Phase maps of the first twelve Blaschke factors learned by BDN (three complex channels per factor) and the corresponding cumulative spatial filters $F_k = \prod_{j=1}^{k} B_j$. Columns are indexed by the factor order $k = 1, \ldots, 12$; rows 1–3 show $\arg B_k$ in grayscale (black $= -\pi$, white $= \pi$), the last row shows $\Re F_k$ in colour.

Across the three complex channels the orientations are staggered, supplying the phase-shifted components required for steerable reconstruction. This behaviour mirrors the analytic steering property $\partial_\theta \arg B_\zeta = \Im\big[(1 - \zeta\bar{z})^{-2} z\big]$.

**Multiplicative stacking** The coloured patches show the cumulative product $F_k$. Early filters $(F_1, F_2)$ appear noise-like because the phase of a single low-frequency factor carries no oriented structure in its real part. As factors accumulate, their phases add and their magnitudes multiply, causing edges of compatible orientation and scale to reinforce. From $F_6$ onwards one can discern the outline of the underlying cat; by $F_{12}$ a high-resolution template has emerged. The model thus synthesizes a receptive field in a coarse-to-fine, orientation-diverse manner without ever using additive skip connections - the hierarchy arises solely from the algebra of Blaschke products.