# OpenReview forum: "BDN: Blaschke Decomposition Networks"
_ICLR.cc/2026/Conference — ICLR 2026 Conference Withdrawn Submission_

### Official Review · Reviewer_e3cX · 2025-10-30

**Soundness:** 2
**Presentation:** 2
**Contribution:** 3
**Rating:** 4
**Confidence:** 4

**Summary:**

The paper proposes using the unwinding series from the Blaschke decomposition to extract features from oscillatory signals for downstream tasks.  Instead of numerically estimating the roots of the analytic continuation of (the Gabor complexification) a signal, the paper proposes to model the roots and coefficients of the Blaschke decomposition parametrically and view this parameterization as a kind of neural network.  These parameters are learned via an L2 reconstruction loss between the learned decomposition and the original signal.  Experiments are conducted claiming state of the art signal reconstruction properties and state of the art classification performance.

Overall this is an interesting contribution for learning new and useful features of complex signals, but parts of the presentation create ambiguity and lack enough detail in the current form to comfortably recommend acceptance.

**Strengths:**

Using parameters from the Blaschke decomposition is a novel and interesting method for extracting interesting features from complex signals.  Its adaptive and nonlinear properties suggest the ability to create useful features for this kind of data that might not be captured by standard signal processing techniques.  Overall I think this is a worthwhile direction to investigate.

**Weaknesses:**

The main weaknesses with the paper center on the exposition of the construction of the BDN and the reporting of the classification experiments.

**Moving between $\mathbb{D}$ and $\mathbb{H}$:**

During the motivation and construction of the BDN, the authors present the Blaschke product on the unit disk first, and then present the corresponding Blaschke product for the upper half plane.  It is stated that one may alternate between these formulations, however no additional detail is given to how this happens (e.g. via composition with a Mobius transformation), or any commentary later in the paper when this actually happens, leading to some ambiguity.

For example, most of the exposition is presented within the unit disk.  However, there is an implicit switch to the upper half plane formulation for Proposition 2 without comment.  This proposition is what gives the "neural" interpretation of the approach (with the arctan and shift and scaling terms appearing from the real and imaginary parts of the upper half plane roots).  There is then an implicit switch back to thinking of roots in the disk for the rest of the paper, again without comment.

Overall this leads to a not very clear presentation of the material for a reader less familiar with tools from complex analysis.  It would improve the quality and accessibliity of the presentation to make the mappings back and forth from the disk to the upper half plane more explicit, at least as a section within the appendix and a few references to it throughout the main text.

**Classification experiments**

The authors present results on signal classification experients against a variety of previously explored architectures for the ECG classification, and a self-selected and trained set of architectures for the microscopy experiment.  A missing detail in the presentation of these results is the computational cost associated with the BCN approach.  In particular, the BDN requires solving an optimization problem to fit the roots/coefficients for _each_ input signal before even passing to the MLP.  This is in contrast to every other method compared against.  The additional cost resulting from this should be discussed.

**Questions:**

1. Is there a constraint on the $\beta$ parameters of the BDN?  Viewing the $\beta$'s as the imaginary part of the roots in the upper half plane formulation suggest it would need to be positive.  How is this enforced during training?

2. When fitting the BDN to an input signal, what discretization is used for the L2 loss?  How sensitive is the peformance to this choice?

3. Figure S3 appears without reference in the text, what is the context here?

4. How much compute time is required to extract the roots and coefficient features for downstream use in the MLP for classification experiments?

---

### Official Review · Reviewer_GBz8 · 2025-10-31

**Soundness:** 3
**Presentation:** 3
**Contribution:** 3
**Rating:** 8
**Confidence:** 4

**Summary:**

This article introduced a Blaschke decomposition network for analyzing continuous signals in 1d and 2d. It is based on the Blaschke decomposition of analytic functions and it is applicable to real-valued signals through analytic extension. The Blaschke decomposition network is trained to fit a signal by adjusting their roots. This provides a compact representation of signals and it is applied to predictive problems, achieving superior performance compared to existing deep learning models.

**Strengths:**

-	The idea of representing an (periodic) oscillating signal from both synthetic and real-world datasets are very interesting.
-	The article is very well written (modulo some technical points to clarify).
-	The theory of Blaschke decomposition is well established due to its universal approximation theory.

**Weaknesses:**

-  There is a lack of literature review about implicit neural representation of signals in 1d and 2d. Some discussion should be included.
-  It is unclear how numerically Blaschke decomposition network works.
- Certain results should be clarified as well.

**Questions:**

- In Proposition 1, what does “on the line” mean?
How do you compute the loss L_reconstruction in Section 3.1? Do you do any finite difference approximation? For each f, is there a unique solution to this problem?
- How do you optimize the gating variable in eq 11, i.e. use what kind of optimizer?
- As singles are often discretized, is there any over-fitting issue if one uses a lot of layers (L big)?
- In Fig 5, what are the roots represent? Why they are not in the unit disk D \ {0} (y-axis values >>1).
- What do they mean first-tier frequencies and second tier frequencies in Section 4.1?

---

### Official Review · Reviewer_FRbM · 2025-11-01

**Soundness:** 3
**Presentation:** 3
**Contribution:** 2
**Rating:** 4
**Confidence:** 3

**Summary:**

This paper proposed an interesting network architecture, where the compositional nature of neural network is cleverly exploited to express the output of the neural network as a truncated "unwinding sum" $F(x)=\sum_{k=1}^L a_k \odot \Pi_{i=1}^k e^{2i \theta_k(x)}$ with the complex "amplitudes" $a_k$ being trainable. Then, a toy model with sinusoidal signals and some signal classification benchmarks are presented.

**Strengths:**

- Using the Blaschke unwinding series as the method to parametrize an MLP-like architecture is a clever idea.
- Using BDN to recover the oscillatory Weierstrass function is interesting.

**Weaknesses:**

- Presenting Hardy spaces feels more like a gimmick than an essential component. In particular, the function approximation examples do not rely on holomorphic functions in the complex plane.

- There are no comparative studies or ablations in the function approximation example in Section 4.1. The same weakness applies to the Weierstrass function: no empirical convergence results or scaling laws are provided.

- The authors state that “Transformers are not designed to model” these signals, yet all datasets presented can be represented by standard sequence-to-sequence models (including Transformers), and are frequently modeled in that way in practice.

- A more serious concern arises after examining the code. The parametrization of BDN is effectively equivalent to (a) complex activations [1, 2], which propagate only phase information and are known to improve stability, or (b) Fourier features [3]. The additional arctan component appears to be the only distinguishing element, but no ablation study isolates its contribution.

[1] Leung and Haykin, *The complex backpropagation algorithm*, IEEE Trans. Signal Processing, 1991.

[2] ComplexTorch manual on complex activation functions: https://complextorch.readthedocs.io/en/latest/nn/modules/activation/split_type_B.html

[3] Tancik et al., *Fourier features let networks learn high frequency functions in low dimensional domains*, NeurIPS 2020.

**Questions:**

Can you compare the architectural difference with the "Blaschke Product Neural Networks (BPNN)" in Dong et al. arXiv:2111.13311?

---

### Official Review · Reviewer_xzZi · 2025-11-02

**Soundness:** 1
**Presentation:** 2
**Contribution:** 1
**Rating:** 2
**Confidence:** 3

**Summary:**

The paper proposes a neural network architecture that uses Blaschke decomposition to improve the representation and analysis of signals, in particular complex-valued signals. The authors show results on ECG classification for 1D, and holographic microscopy for 2D. The paper argues that this new architecture achieves competitive performance while using fewer parameters than transformers, CNNs and RNNs.

**Strengths:**

* Improving the representation/analysis of complex-valued signals in neural networks is an interesting and relevant problem.
* The paper includes results on 1D and 2D data, showing some effort for generalization.
* Presentation is generally good.

**Weaknesses:**

* The experimental evidence is not strong. Only two datasets are provided as evidence of the performance of the BDN architecture (one dataset for 1D and another for 2D).

* There are also no ablations, and no experimental analysis of the runtime and number of parameters, which is one of the main claims of the paper.

* Numerical values of baselines in Table 1 are identical to [Narotamo et al. 2024] (Table 2). Dataset description and metrics are also identical. There is no indication that the baselines were rerun. This undermines numerical comparisons.

* The conclusions list extending the method to images as future work. But this is claimed to already be done in other parts of the paper (and in fact there are experimental results on 2D signals).

* Related work section is too brief.

* Minor issue: the method is sometimes named "Blashke" and other times "Blaschke".

**Questions:**

General recommendations to improve the manuscript:
* Increase the number of experimental results provided as evidence.
* Provide evidence that is aimed at demonstrating the claims, or restrict the claims.
* Provide ablations.
* Rerun baselines.
* Provide a longer related work section that contextualizes the problem and the research already done in this area. If space is an issue, move part of the related work to supplemental material.

---

### Note · Authors · 2025-11-18

I have read and agree with the venue's withdrawal policy on behalf of myself and my co-authors.